# Vertical Jump Tests: A Safe Instrument to Improve the Accuracy of the Functional Capacity Assessment in Robust Older Women

**DOI:** 10.3390/healthcare10020323

**Published:** 2022-02-08

**Authors:** Carlos A. F. Santos, Gislene R. Amirato, Alessandro F. Jacinto, Ana V. Pedrosa, Adriana Caldo-Silva, António R. Sampaio, Nuno Pimenta, Juliana M. B. Santos, Alberto Pochini, André L. L. Bachi

**Affiliations:** 1Discipline of Geriatrics and Gerontology, Federal University of Sao Paulo (UNIFESP), Sao Paulo 04020-050, Brazil; alessandrojacinto@uol.com.br; 2Postgraduate Program in Health Science Applied to Sports and Physical Activity, Paulista School of Medicine (EPM), Sao Paulo 04022-001, Brazil; apochini@uol.com.br; 3Mane Garrincha Sports Education Center, Sports Department of the Municipality of Sao Paulo (SEME), Sao Paulo 04039-034, Brazil; prof.gislene@hotmail.com; 4Postgraduate Program in Translational Medicine, Federal University of Sao Paulo (UNIFESP), Sao Paulo 04039-002, Brazil; 5Research Unit for Sport and Physical Activity (CIDAF), Faculty of Sport Science and Physical Education, (FCDEF-UC), University of Coimbra, 3040-248 Coimbra, Portugal; anapedrosa90@gmail.com (A.V.P.); dricaldo@gmail.com (A.C.-S.); 6N2i, Polytechnic Institute of Maia, 4475-690 Maia, Portugal; arsampaio@ismai.pt (A.R.S.); d011557@ipmaia.pt (N.P.); 7Post-Graduation Program in Science of Human Movement and Rehabilitation, Federal University of Sao Paulo (UNIFESP), Santos 11015-020, Brazil; juliana-mbs@hotmail.com; 8Post-Graduation Program in Health Sciences, Santo Amaro University (UNISA), Sao Paulo 04829-300, Brazil; allbachi77@gmail.com

**Keywords:** aging, vertical jump, isokinetic, power, strength, muscle, function tests

## Abstract

Age-related decreases in muscle function lead to disabilities and are associated with negative health outcomes in older people. Although several physical tests can be used to assess physical performance, muscle strength, and power, their interpretation can be hampered by the ceiling effect of some of them. The aim of this study was to assess whether vertical jump tests are safe in terms of physical integrity and whether they are useful in assessing physical performance in forty-one robust older women. The investigation entailed an assessment of anthropometric characteristics, physical functioning tests (Short Physical Performance Battery (SPPB), sit-to-up 5 times and sit-to-up 30 s, gait speed, time-up-to-go test (TUGT)), and tests evaluating muscle strength and power (handgrip, lower limb isokinetic tests, and vertical jumping tests). Significant negative correlations were found between vertical jumping tests and BMI, body fat percentage, sit-to-up 5 times and TUGT. In addition, significant positive correlations were observed between vertical jumping tests and SPPB, gait speed, handgrip, and concentric isokinetic tests of knee muscles. No adverse events in volunteers’ physical integrity were reported during and after the performance of all physical tests. Thus, the study results showed that vertical jumping tests are safe and accurate for assessing physical performance and are useful for monitoring age-related loss of muscle performance in robust older women.

## 1. Introduction

The aging process is a natural phenomenon that, among some aspects, is characterized by a heterogeneous loss of muscle performance that begins long before the individual becomes aged. It is estimated that between the fourth and seventh decades of life there is a 50% reduction in human muscle mass and strength [1], mainly due to the reduction of type II muscle fibers [2]. However, in agreement with the scientific literature, muscle power is reduced prior to the loss of muscle strength during aging [3]. In this sense, it is paramount to highlight, particularly from the age of seventy onward, that the loss of muscle function is accelerated, with an evident reduction in power over muscle strength, probably due to increased fatty infiltration [4]. However, these physiological changes are not capable of causing disabilities in the basic and instrumental functions of aged people [5].

It is widely accepted that some age-related aspects, such as immunosenescence, inflamm-aging, lifestyle and habits, nutrition, polypharmacy, hormonal issues, and presence of chronic diseases, can influence and increase the vulnerability to develop diseases and geriatric syndromes associated with physical disabilities and functional dependence, preferentially sarcopenia and frailty [6,7]. Based on these facts, the World Health Organization (WHO) Clinical Consortium on Healthy Ageing (CCHA), held in November 2019, was the fifth gathering of an international group of clinical leaders, in which one of the objectives was to create strategies to reduce the number of people dependent on care by 15 million by 2025 [8]. However, in accordance with recent reports, this number will probably need to be revised because of the pandemic, originating from SARS-CoV-2, which has resulted in an increase in the incidence of frailty syndrome and sarcopenia for several reasons: acute infectious disease, social isolation, lack of follow-up for chronic diseases, and failure to diagnose new chronic diseases [9,10].

In terms of assessment of muscle and physical function in aging, it is known that there are many valuable instruments and tests that not only allow the verification of the ability of older adults to perform typical day-to-day tasks but also that can assist in identifying vulnerabilities for the development of frailty syndrome and diagnosis of sarcopenia. Incidentally, the most commonly used simple physical tests are: the handgrip (HG), timed up-and-go test (TUGT), multiple sit-to-stand field test, gait speed, and the Short Physical Performance Battery (SPPB) [11,12]. Except for HG, which evaluates the maximum force applied in a handheld dynamometer, the other tests require a high degree of coordination from different muscle groups, are low intensity (in terms of using maximal muscle force or power), and rely heavily on other body organs such as the eyes, vestibular system, and proprioception in order to be performed [13]. Although these tests are many applied, they present limitations, for instance: (1) HG only measures the performance of the muscle group related to the hands and wrists; (2) TUGT, gait speed, the multiple sit-to-stand may have their results influenced by the examiner; and (3) the presence of a ceiling effect is observed when these tests are performed by robust older adults [14,15].

Regarding the term “robust”, it refers to older adults who have autonomy and physical independence, with healthy lifestyle habits and low vulnerability to geriatric syndromes such as dementia and frailty [16,17]. Although they can be considered as strong individuals, it is important to clarify that the healthy independent seniors of today may become the sarcopenic or frail seniors of tomorrow. Thus, there is a growing need to better understand not only the epidemiology, but the course of muscle function loss and its associations with other clinical outcomes as well [18]. In this respect, the tests formerly described can present low sensitivity to detect alterations over time in robust older adults, which requires the use of more sensitive and reproducible tools [19].

Beyond the instruments and physical tests frequently used in older adults, the isokinetic tests of the knee flexor and extensor muscles and the contact platform vertical jump tests are two possible evaluations of lower limb muscle strength and power [20,21]. Particularly, these tests are digitized, and there is no interference from the examiner. In relation to the isokinetic test, it requires a specific dynamometer, which is found in research laboratories, rehabilitation clinics, and high-performance sports clubs. Its application can be performed in several muscle groups, in an isolated way, in a closed chain, and at different speeds in which muscle strength or power can be evaluated [22]. Concerning the vertical jump tests, which can be performed on a contact platform set up stably at ground level, they are generally found in rehabilitation clinics and gyms. In terms of the mechanics of the vertical jump, it is necessary for the application of a force perpendicular to the ground and contrary to the action of gravity. Besides, it is needed for balance and proprioception at the time of departure from and time of landing on the ground. The performance of the vertical jump is related to the physical concept of muscular power: a great force applied in a small interval of time [23].

In both tests, the examiner should encourage the person being evaluated to exert as much effort as possible since the purpose of these tests is to assess maximum strength and muscle power. Therefore, care and safety with regard to physical integrity during the familiarization and performance of these tools are essential, especially in the vertical jump test, which is an open-chain activity with risk of falling and joint damage during the landing phase [24,25]. Previous studies have shown that vertical jump tests are feasible and can be performed safely, without the occurrence of falls, vertebral fractures, or accidents during their performance, even in older people with osteoporosis [11,23,24,25,26].

Based on these data, the aim of the present study was to determine the safety, in terms of physical integrity, and accuracy of vertical jump tests in a group of older robust women, and to compare results against traditional function and isokinetic tests.

## 2. Materials and Methods

Initially, 49 women aged ≥60 years were invited to participate voluntarily in this cross-sectional study between March and April 2017. Volunteers were recruited from the geriatric outpatient service of the Sports Medicine Discipline/UNIFESP, and all of them were participating in a regular program of combined exercise training (aerobic and resistance) at moderate intensity under supervision and guidance of physical educators at Mané Garrincha Sports Education Center (Sports Department of São Paulo City) for at least 1 year. Study inclusion criteria were: (1) no contraindication for engaging in a moderate-intensity exercise training program, based on clinical evaluation and medical record review; (2) no dementia syndromes; (3) signing of the informed consent form. The exclusion criterion was presenting pain or physical discomfort during the days of the physical assessments. All of the volunteers signed the informed consent form (TCLE) previously approved by the Ethics and Research Committee of the Federal University of São Paulo (UNIFESP), under number 0692/2017.

As presented in the flow diagram (Figure 1), 4 volunteers were excluded for clinical reasons, and the other 4 volunteers were excluded for orthopedic issues. Therefore, 41 older-aged women, with a mean age of 71 ± 6.2 years, were enrolled in the present study since they had met the eligibility criteria and had completed all the evaluations purposed.

The volunteers were evaluated at the geriatric outpatient service in the Discipline of Sports Medicine/UNIFESP by a geriatric physician who applied function tests, described below, for physical performance and who measured participant body weight using a digital scale (Personal^®^ scale, Filizzola, São Paulo, Brazil) accurately to the nearest 0.1 kg. Body height was measured using a wall-mounted stadiometer, accurate to the nearest 0.1 cm. The women wore light clothes and no footwear. Body mass index (BMI) was calculated by weight over height squared (kg/m^2^). The left calf circumference was determined by using a measuring tape to the nearest 0.1 cm with the volunteer seated, knees bent at 90 degrees and feet supported. For the tests that needed time measurements, a stopwatch was used (I-phone 5^®^ Apple Inc., Cupertino, CA, USA). In relation to the application of simple tests of physical performance, we used traditional protocols described in the scientific literature [12,13,27,28]. Results were expressed in seconds for the timed up-and-go test (TUGT) and for the sit-to-stand test in the chair for 5 repetitions (SIT-UP 5X); in meters/second for the gait speed (GS) test; in number of repetitions for the sit-to-stand test in the chair for thirty seconds (SIT-UP 30”); in points for the Short Performance Physical Battery (SPPB); and in kilograms of force for the handgrip (HG) Cupertino. Regarding the handgrip test, the best performance out of three attempts, with a 1 min interval, using the dominant hand was recorded in position two on an analog dynamometer (Jamar Hydraulic Hand Dynamometer^®^, Sammons Preston Rolyan, Bollingbrook, IL, USA). All of these assessments were performed by the same medical geriatrician.

In relation to the performance of the other physical tests (isokinetic tests of muscle strength and power and vertical jump tests) and the assessment of body composition, a second meeting was scheduled one week after the first, during which the familiarization of these activities was performed. Regarding familiarization with the isokinetic tests, it was carried out in two stages: first, in the week before the test, each volunteer accessed the equipment and received theoretical training about the test, and on the day of the test, all of them performed some movements before the beginning of the test. Concerning familiarization with the vertical jump tests, it was carried out in the week before the tests, by jump training, which occurred during their gym class.

According to the protocol described by Neder and collaborators [20], each volunteer was oriented to walk for 5 min before the test performance in order to activate the muscles of the lower limbs that were to be required in the isokinetic tests. To perform this test, we used the equipment isokinetic Biodex (Multi-Joint System 3^®^ digital dynamometer, Biodex Medical Inc., Shirley, NY, USA). The peak torque of the isokinetic test of concentric contraction of the extensor (Ext) and flexor (Flex) muscles of the knee at 60 and 180 degrees per second (60°·s^−1^ and 180°·s^−1^) was used in the evaluation of muscle strength and power. The results were expressed in Newton-meters per kilogram of body weight (Nm·kg^−1^). The tests were performed in the seated position with 85 degrees of hip flexion. The dynamometer lever arm was positioned parallel to the patient’s leg, with the resistance pad was attached directly above the lateral malleolus of the fibula. The joint axis of the knee was aligned with the rotational axis of the dynamometer by an imaginary line passing through the lateral femoral epicondyle. Velcro bands were used to stabilize the trunk, hip and lower limbs evaluated. The subject’s hands were supported on the sides of the equipment. In the initial position, the neutral position was defined with maximum knee flexion. Prior to the test, calibration procedures and correction of gravity of the isokinetic dynamometer were adopted according to the manufacturer’s instructions (Biodex Medical Inc.^®^). At the beginning of the evaluation, the volunteers performed three submaximal repetitions to become familiar with the equipment. After familiarization, the volunteers performed five maximal repetitions in concentric action at the chosen angular velocity. The tests were performed on both legs, starting with the dominant limb. The first test was performed at an angular velocity of 60°·s^−1^ and, after a 15 min rest, the test was performed again at an angular velocity of 180°·s^−1^. Phrases of encouragement were used to motivate the volunteers.

Regarding the vertical jump test assessment, it was performed on a jumping platform (Elite Jump^®^, S2 Sports, São Paulo, Brazil), and digital results were expressed in centimeters, representing jump height. In accordance with Loturco and collaborators [21], each volunteer was oriented to walk for 5 min before performing the jump tests in order to activate muscles of the lower limbs that were to be required in this test. Two types of evaluations were carried out to measure the height of vertical jumps: the first was countermovement jumps (CMJ), followed by squat jumps (SJ). All volunteers were submitted to familiarization with these exercises in the week preceding the evaluations, following the protocol formerly described. For CMJ, volunteers placed their hands on their hips and were instructed to perform a downward movement followed by complete extension of the legs, being free to determine the range of the countermovement through motor coordination. For SJ, volunteers were required to remain static in a knee flexion position in a comfortable, self-selected position for two seconds before the jump, without any preparatory movement. Five jumps of each style were performed, with intervals of 15 s between them. A 5 min break was given between CMJ and SJ styles. The highest jump for CMS and SJ was used for the analysis [21].

The evaluation of body composition using dual-emission X-ray absorptiometry (DEXA, version software 12.3, Lunar DPX, GE Health, Madison, WI, USA) was performed at the second meeting during the time interval between the isokinetic and vertical jump tests. For the DEXA evaluation, participants were instructed to wear underwear only. By using the device software, the results of skeletal muscle mass by segment and percentage of total body fat for each volunteer was obtained. The appendicular skeletal mass index (ASMi) was used, calculated as appendicular skeletal muscle mass (ASM) (kg) of the four limbs divided by height squared (ASM/m^2^) [11].

During the performance of the tests, to assess safety in a context of physical integrity, the occurrence or lack of falls, as well as the presence or absence of pain, physical incidents were assessed. In addition, one week after the second meeting, we contacted all volunteers by telephone call to provide feedback concerning the results of the assessments and to verify the occurrence of any discomfort or pain that had begun after the physical tests were performed.

First, all data were evaluated for normality using the Shapiro–Wilk test, confirming that these were parametric variables. Therefore, results were expressed as mean and standard deviation (X ± SD). Pearson’s correlation coefficient was used to evaluate the association between the parameters assessed in the study. Significance level was set at 5% (*p* < 0.05).

## 3. Results

Table 1 shows not only the anthropometric characteristics of older women enrolled in the present study but also Pearson’s correlation coefficient analysis of anthropometric measurements and results from all physical performance tests. Positive correlations were found between BMI and calf circumference, total body fat percentage, appendicular skeletal mass index, and peak torque (60°·s^−1^ knee extensor), including between calf circumference and total body fat percentage. However, the countermovement jump test showed a negative correlation with total body fat percentage, and the squat jump test showed negative correlations with BMI, calf circumference, and total body fat percentage.

Pearson’s correlation coefficient analysis of the traditional physical performance tests (function tests) applied in the present study is shown in Table 2. In relation to the analysis of function tests, as expected, positive correlations were observed between the TUGT and sit-to-up 5 times test, as well as between the SPPB and sit-to-up 30 s and handgrip tests. In addition, negative correlations were found not only between the TUGT and gait speed, sit-to-up 30 s, SPPB, and handgrip tests but also between sit-to-up 5 times and SPPB and sit-to-up 30 s tests.

Results of the Pearson’s correlation coefficient analysis of the traditional physical performance tests and vertical jump tests are depicted in Figure 2. First, negative correlations between both vertical jump tests applied in this study and sit-to-up 5 times (C and J) were observed, whereas only the countermovement jump showed a negative correlation with the TUGT (F). In addition, positive correlations were found between both vertical jump tests and SPPB (A and I), sit-to-up 30 s (D and K), handgrip (E and L), and peak torque 60°·s^−1^ knee extensor (G and M), whereas the countermovement jump also showed positive correlations with gait speed (B) and peak torque 60°·s^−1^ knee flexor (H).

Results of Pearson’s correlation coefficient analysis of isokinetic tests of knee muscles at 60°·s^−1^ and physical function tests are depicted in Figure 3. Positive correlations between isokinetic tests (peak torque of extensor and flexor knee muscles) and SPPB (A and D), gait speed (B and E), and handgrip (C and F) were found. In addition, as expected, a positive correlation was found between peak torque of extensor and flexor knee muscles (G).

Concerning the evaluation of the volunteers’ physical safety in this study, falls and physical accidents during their performances were not verified. In addition, contact was made by telephone one week following performance test, and we verified that there was no occurrence of discomfort or pain related to the isokinetic evaluation and vertical jump tests.

## 4. Discussion

Our results showed that, generally, vertical jump tests are safe tools since they did not cause alterations in the physical integrity of the group of robust older women, and these tests were significantly correlated with both traditional physical performance tests and anthropometric measures. More specifically, the study findings confirmed that: (a) vertical jump tests, in a similar way to the sit-to-up test for 30 s, showed strong negative correlations with BMI and total body fat percentage; (b) vertical jump tests, similar to isokinetic tests (60°·s^−1^ extensor and flexor), showed strong positive correlations with the handgrip test; (c) vertical jump tests showed a positive correlation with isokinetic tests, but only for 60°·s^−1^ peak torque; and (d) vertical jump tests showed both positive (7) and negative (3) correlations with functional physical tests applied in this study.

Regarding the functional correlations, it is notable that the total number (10) of correlations observed between vertical jump and functioning tests was higher than the total number (six) between isokinetic and function tests. In general, the performance of function tests involves the physical ability related to balance, movement speed, submaximal muscle strength and power. In the context of aging, all these characteristics are related to the performance of basic and instrumental activities in daily living [13,29,30,31]. Thus, this finding allows us to putatively suggest that vertical jump tests can also be applied to assess not only the functional aspects, such as muscle strength and power, but also the daily activities in a group of robust older women.

As previously reported, functional tests have lower specificity and sensitivity for assessing losses in muscle strength and power, particularly among robust older people [14,15,16]. Based on this premise, only robust older women were included in the present study in order to determine whether the utility of vertical jump tests could be extended to measure alterations in muscle function. All volunteers enrolled in this study were engaged in an exercise training program and had excellent adherence to the orientations concerning the healthy habits provided by the outpatient service for Sports Medicine and Geriatrics. Furthermore, the physical performance and body composition results of all participants demonstrated that they did not present sarcopenia [7,18].

It is known that during aging, there is typically a loss of maximum strength in the lower limbs. However, the reduction in muscle power is the earliest and most important factor for the decline in physical performance. Taken together, these musculoskeletal changes are age related neuromechanical parameters and can represent significant predictors of mobility decline, sarcopenia, and frailty in older adults [6,18,32]. 

Therefore, in this study, maximum muscle strength and power performance of the lower limbs were determined using isokinetic assessments and jumping tests. Concentric isokinetic tests performed at speeds of 60°·s^−1^ (low speed) and 180°·s^−1^ (high speed) are associated with strength and power of the knee extensor and flexor muscles, respectively. Similarly, for the execution of the vertical jump tests, strength and power of the lower limbs are necessary; it requires the application of force to generate the propulsion of body mass vertically against the action of gravity, whereas the squat jump and countermovement jump styles require balance, isometric and isokinetic strength, and explosive movements to execute [21,33,34,35,36]. We verified that the results showed a significant positive correlation between the vertical jump tests and isokinetic tests, specifically in terms of peak torque at low speeds (60°·s^−1^) yet not at high speeds (180°·s^−1^). It is of utmost importance that we did not find studies that presented results demonstrating associations (positive or negative) between isokinetic tests in high speed or vertical jump tests. However, the lack of correlation at high speeds can be explained by the fact that this population of older women was already experiencing age-related loss in muscle power because, although they are robust, these women suffer from the physiological process of aging, related to the reduction of muscle power even before losing strength [2,3]. Thus, we can suggest that the volunteer participants in this study exhibited a lesser ability to perform the movements at high speed, which is not habitual for them but which is required in the isokinetic test (180°·s^−1^).

Besides the correlations cited above, the vertical jump tests also showed good positive correlations with the handgrip test, which assesses the maximum concentric strength of the wrist and hand muscles. The handgrip test, numbering among the traditional tests of function, is widely used to assess strength and vulnerability to disability, allowing the diagnosis of dynapenia, sarcopenia, and frailty in older people. These findings also corroborate the capacity of vertical jump tests to assess muscle strength, being an alternative to handgrip when the outcomes are related to mobility changes [17,26,37,38,39,40].

As expected, performance during the vertical jump tests also correlated with best performance during the sit-to-up 30 s test and the 5 times sit-to-up test, since the neuromuscular mechanics of the movement of standing up and sitting in a chair are similar to those of jumping [41,42,43]. In addition, only the vertical jump tests exhibited significant negative correlations with BMI and total body fat percentage. These findings corroborate the recent report by Moore and collaborators [44], in which the performance of the jump test was negatively associated with fat mass in older adults. Thus, our observations can putatively allow us to propose that vertical jump tests are more sensitive than the sit-to-up 30 s test, the handgrip test, and the isokinetic test for discriminating the influence of body fat on muscle strength in aged robust women. It is possible to imagine that the negative function of fat is more evident in the application of the power of the vertical jump [4,45].

Beyond the fact that vertical jump tests prove to be a good option for assessing muscle strength and power and are similar to reports in other studies [23,24,25,26,33], in this study, no clinical adverse events resulting from the tests applied were evidenced. Therefore, based on these data, the vertical jump tests showed safety, feasibility, reliability, and reproducibility, including for older adults.

## 5. Limitations of the Study

One limitation of this study is related to the eligibility criteria, since to perform the study, it was advocated toward older women against a profile of active people, practitioners of physical exercises, and those accustomed to performing vertical jumps. Thus, the volunteers enrolled in this study could be considered as robust aged women because they demonstrated physical performance well above the established limits for negative musculoskeletal outcomes. Thus, these facts can be characterized as a sample bias due to these volunteers probably being more prepared to jump with a lower risk of falls and accidents. Moreover, another limitation of this study is associated with the fact that this is a cross-sectional study, and cause–effect associations could not be evaluated.

## 6. Conclusions

Based on our results, vertical jump tests were safe, in terms of maintaining physical integrity, and accurate for assessing physical performance in robust older women since they showed significant correlation with the traditional functional tests. Therefore, we can suggest that vertical jump tests can be a valuable tool to monitor age-associated loss of muscle strength and power, particularly in lower limbs, as well as to evaluate the performance gain in muscle strength and power in a physical exercise program.

## Figures and Tables

**Figure 1 healthcare-10-00323-f001:**
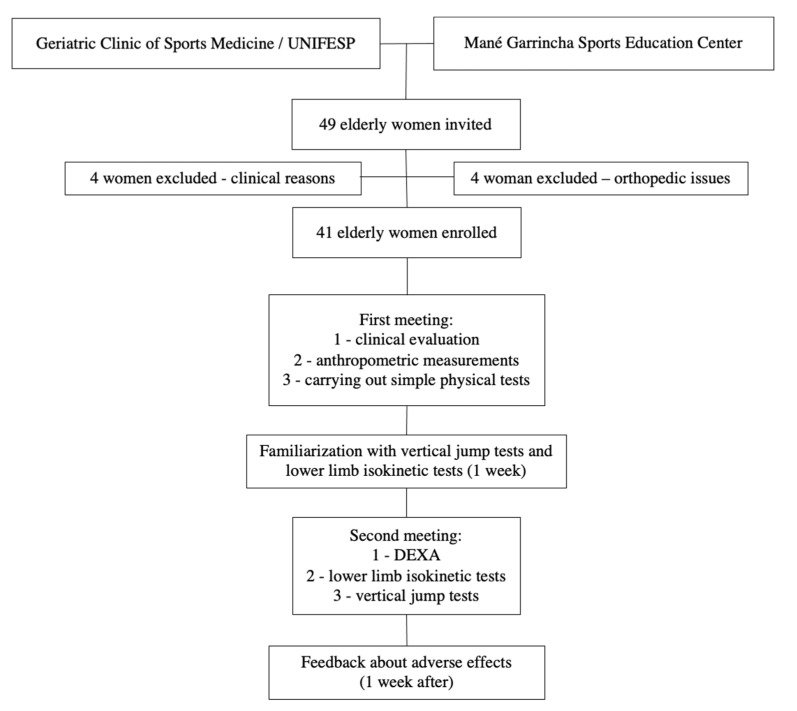
Flow diagram and study design.

**Figure 2 healthcare-10-00323-f002:**
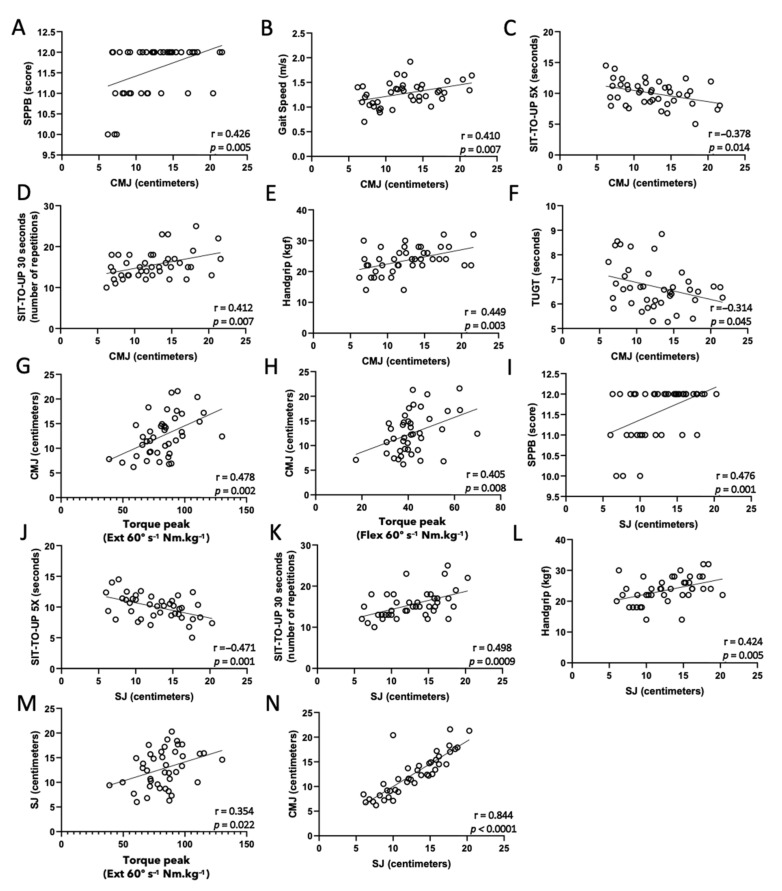
Pearson’s coefficient correlation analysis of vertical jump tests in two styles: countermovement jump (CMJ, (**A**–**H**)) and squat jump (SJ, (**I**–**M**)); physical function tests (SPPB, (**A**,**I**); gait speed, (**B**); sit-to-up 5X, (**C**,**J**); sit-to-up 30 s, (**D**,**K**); handgrip, (**E**,**L**); TUGT, (**F**)). Isokinetic tests: knee muscles at 60 degrees per second (extensor, (**G**,**M**); flexor, (**H**)); between CMJ and SJ (**N**). Significance level: *p* < 0.05.

**Figure 3 healthcare-10-00323-f003:**
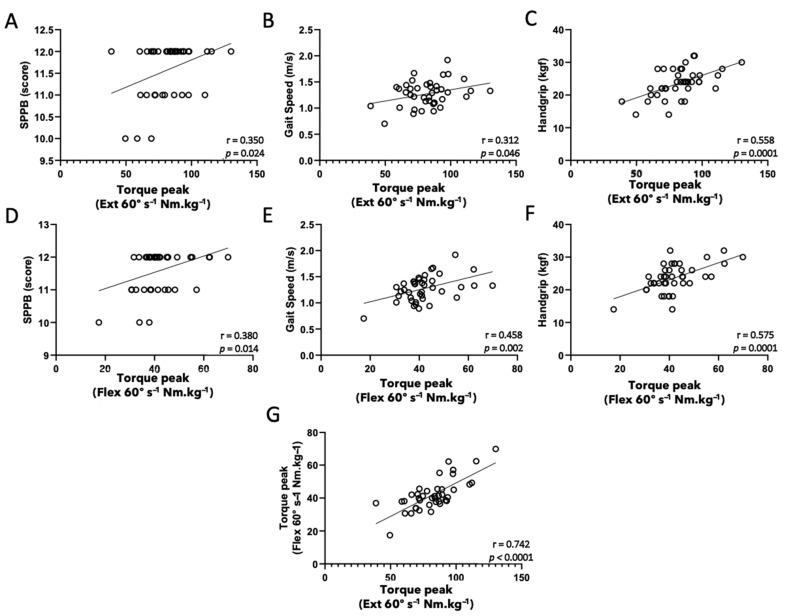
Pearson’s coefficient correlation analysis of isokinetic tests of knee muscles at 60 degrees per second (extensor, (**A**–**C**); flexor, (**D**–**F**)) and physical function tests (SPPB, (**A**,**D**); gait speed, (**B**) and handgrip, (**C**,**F**)); between extensor and flexor knee muscles (**G**). Significance level: *p* < 0.05.

**Table 1 healthcare-10-00323-t001:** Mean and standard deviation of anthropometric characteristics (weight, height, BMI, CALF, % FAT, and ASMi), and physical performance tests of the volunteers enrolled in this study. In addition, Pearson’s coefficient correlation analysis of anthropometrics data and physical performance tests. Significance level *p* < 0.05.

Parameter	Mean ± SD	BMI		CALF		% FAT		ASMi	
		r	*p* Value	r	*p* Value	r	*p* Value	r	*p* Value
Weight (kg)	60.7 ± 10.9	1	n.a.	1	n.a.	1	n.a.	1	n.a.
Height (m)	1.55 ± 0,1	1	n.a.	1	n.a.	1	n.a.	1	n.a.
BMI (kg/m^2^)	25.3 ± 3.9	1	n.a.	1	n.a.	1	n.a.	1	n.a.
CALF (cm)	34.3 ± 2.9	0.551	<0.0001	1	n.a.	1	n.a.	1	n.a.
% FAT	38.9 ± 7.7	0.789	<0.0001	0.453	0.003	1	n.a.	1	n.a.
ASMi (g/m^2^)	6.5 ± 0.6	0.315	0.045	0.163	0.310	−0.125	0.438	1	n.a.
SIT-UP 5X (s)	10.0 ± 2.0	0.232	0.144	0.275	0.082	0.256	0.106	0.051	0.752
GS (m/s)	1.5 ± 0.2	−0.221	0.165	−0.063	0.695	−0.267	0.092	−0.017	0.915
SPPB (score)	11.6 ± 0.6	−0.184	0.249	−0.204	0.200	−0.229	0.149	−0.112	0.485
SIT-UP 30” (repetitions)	16.0 ± 3.3	−0.273	0.084	−0.303	0.054	−0.282	0.074	−0.006	0.970
HG (kgf)	23.6 ± 4.3	0.139	0.385	0.026	0.870	0.064	0.692	−0.001	0.993
TUGT (s)	6.7 ± 0.9	0.053	0.741	−0.089	0.581	0.028	0.864	0.157	0.328
CMJ (cm)	12.5 ± 6.2	−0.236	0.138	−0.253	0.110	−0.349	0.025	0.185	0.247
SJ (cm)	12.5 ± 6.2	−0.313	0.046	−0.373	0.016	−0.425	0.006	0.258	0.103
Ext 60°·s^−1^ (Nm·kg^−1^)	82.4 ± 17.4	0.361	0.021	0.191	0.232	0.199	0.213	0.242	0.127
Flex 60°·s^−1^ (Nm·kg^−1^)	42.0 ± 9.5	0.178	0.265	0.096	0.549	−0.059	0.715	0.156	0.329
Ext 180°·s^−1^ (Nm·kg−1)	87.1 ± 14.6	0.112	0.497	−0.285	0.078	0.023	0.888	0.194	0.237
Flex 180°·s^−1^ (Nm·kg^−1^)	49.8 ± 14.7	0.139	0.398	−0.254	0.119	0.098	0.551	0.033	0.844

Note: BMI, body mass index; Calf, calf circumference; % Fat, percentage body fat; ASMi, appendicular skeletal mass index; GS, gait speed; SIT-UP 5X, sit-to-stand test in chair for 5 repetitions; SIT-UP 30”, sit-to-stand test in chair for thirty seconds; SPPB, Short Performance Physical Battery; HG, handgrip; TUGT, timed up-and-go test; SJ, squat jump; CMJ, countermovement jump; Ext 60°·s^−1^, peak torque of isokinetic test of concentric contraction of knee extensor muscle at 60°·s^−1^; Flex 60°·s^−1^, peak torque of isokinetic test of concentric contraction of knee flexor muscle at 60·s^−1^; Ext 180°·s^−1^, peak torque of isokinetic test of concentric contraction of knee extensor muscle at 180°·s^−1^; Flex 180°·s^−1^, peak torque of isokinetic test of concentric contraction of knee flexor muscle at 180°·s^−1^; n.a., not applicable.

**Table 2 healthcare-10-00323-t002:** Pearson’s coefficient correlation analysis of physical functioning tests. Significance level *p* < 0.05.

Parameter	SIT-UP 5X		GS		SPPB		SIT-UP 30”	
	r	*p* Value	r	*p* Value	r	*p* Value	r	*p* Value
GS	−0.072	0.654	1	n.a.	1	n.a.	1	n.a.
SPPB	−0.690	<0.0001	0.222	0.163	1	n.a.	1	n.a.
SIT-UP 30”	−0.915	<0.0001	0.074	0.647	0.541	<0.0001	1	n.a.
HG	−0.298	0.058	0.261	0.100	0.432	0.005	0.236	0.137
TUGT	0.387	0.013	−0.403	0.009	−0.423	0.006	−0.311	0.048

Note: GS, gait speed; SPPB, Short Performance Physical Battery; SIT-UP 30”, sit-to-stand test in chair for thirty seconds; HG, handgrip; TUGT, timed up-and-go test; SIT-UP 5X, sit-to-stand test in chair for 5 repetitions; n.a., not applicable.

## Data Availability

The data for this study can be found in the medical records of the Geriatric Clinic of the Discipline of Sports Medicine at the Federal University of São Paulo.

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
