# Peer review of "Vertical Jump Tests: A Safe Instrument to Improve the Accuracy of the Functional Capacity Assessment in Robust Older Women"

_healthcare, 2022, doi:10.3390/healthcare10020323_

Round 1

Reviewer 1 Report

My recommendations are:
  In the introduction, make many references to the pandemic period and according to section 2, the study took place in 2017, before the pandemic. Accordingly, I recommend that the Introduction section be completely revised.
Lines 264-271 repeat the idea presented in the Methods section. I recommend that you focus on the goal.
I recommend that you mention the limitations of this study and possibly your strengths.
I recommend that the conclusions section be extended.

Author Response

Comments of the Reviewer #1

My recommendations are:

1) In the introduction, make many references to the pandemic period and according to section 2, the study took place in 2017, before the pandemic. Accordingly, I recommend that the Introduction section be completely revised.

Authors´ response: First of all, we would like to thank you for the comment and revision of the study.

In response to your recommendation, we would like to inform you that the "Introduction section" was revised and rewritten in order to focus on the objective of the study. In addition, we also would like to clarify that pieces of information related to the current pandemic situation were altered to improve its meaning in the study context.

2) Lines 264-271 repeat the idea presented in the Methods section. I recommend that you focus on the goal.

Authors´ response: We would like to thank you for the comment and to clarify that the data in these lines were revised and rewritten in order to describe, in a general way, the main results obtained in this study, maintained the focus on the goal of the study, as presented below:

"Our results showed that, in a general way, vertical jump tests could be used safely in a group of older women robust and also that these tests significantly correlated with both traditional physical performance tests and anthropometric measures. More specifically, the study findings confirmed that: (a) vertical jump tests, in a similar way to the sit-to-up test for 30 seconds, showed strong negative correlations with BMI and total body fat percentage; (b) vertical jump tests, similarly to isokinetic tests (60°.s-1 extensor and flexor), showed strong positive correlations with the handgrip test; (c) vertical jump tests showed a positive correlation with isokinetic tests, but only for 60°.s-1 peak torque; and (d) vertical jump tests showed several positive (7) and negative (3) correlations with tests of functioning applied in this study."

3) I recommend that you mention the limitations of this study and possibly your strengths.

Authors´ response: We would like to thank you for the comment and to inform you that a subsection named "Limitations of the study" was added as the last paragraph of the "Discussion section". Below, we presented this paragraph added in the main text.

"One limitation of this study is related to the eligibility criterion since to perform the study it was avocated older women with a profile of active people, practitioners of physical exercises and accustomed to performing vertical jumps. So, the volunteers enrolled in this study could be considered as robust aged women, because they demonstrated physical performance well above the established limits for negative musculoskeletal outcomes. Thus, these facts can be characterized as a sample bias due to these volunteers are probably more prepared to jump with a lower risk of falls and accidents. Moreover, another limitation of this study is associated with the fact that this is a cross-sectional study, and cause/effect associations could not be evaluated."

4) I recommend that the conclusions section be extended.

Authors´ response: We would like to thank you for the comment. As recommended the "Conclusions section" was revised and extended, as presented below:

"Based on our results, vertical jump tests are safe and accurate for assessing physical performance in robust older women since they showed significant correlation with the traditional functional tests. Therefore, we can suggest that vertical jump tests could be a valuable tool not only to monitor age-associated loss of muscle strength and power, particularly in lower limbs, as well as to evaluate the performance gain in muscle strength and power in a physical exercise program. "

Reviewer 2 Report

I attached my comments in a PDF document.

Author Response

Comments of the Reviewer #2

1- introduction: I think that the introduction should be rewritten since it is not completely related to the sample and theme of the work. It focuses on sarcopenia and the frailest but the work does not have this sample. I believe that the potential of the work is the fact of being able to include new tests in a population of older adults that is not fragile and for which we have no data. Yes, mention is made of the ceiling effect, but the idea is not promoted. The study sample seems to have physical function values above the cut-off points and also practice physical activity, therefore, they cannot be assessed under the standards to which we are accustomed since this ceiling effect occurs. Why talk about sarcopenia and other diseases associated with age when it is not the target population of this study.

Authors´ response: First of all, we would like to thank you for the review and comment concerning the present study.

In response to your recommendation, we would like to inform you that the "Introduction section" was revised and rewritten in order to highlight the main characteristics regarding the robust and trained older adults since they were the population that participated in the present study. In addition, we also added pieces of information concerning the importance of applying different ways to assess the physical and functional functions in vulnerable or not older adults, particularly sarcopenic and frail.

2- Results and discussion It could be interesting to include the muscle mass value in the associations, since it will be more related to strength than the fat mass value. I do not think that the fact that there is a negative association between BMI, fat and vertical jumps is the reason why these tests can be more similar to discriminate the influence of body fat on strength, especially without considering mass total muscle and / or limbs. On the other hand, the associations are not cause-effect relationships and although there is an association between handgrip and vertical jump, perhaps another type of study would be needed to raise the possibility that one test can replace another.

Authors´ response: We would like to thank you for the comment and to inform you that, as recommended, we performed a new round of analysis including specifically the appendicular skeletal mass, which represents the free-fat muscle mass of limbs, and was observed the same results when we used the appendicular skeletal mass index (ASMi). In addition, it is noteworthy to clarify that the ASMi (appendicular skeletal mass index or Baumgartner index) represents the relationship between appendicular muscle mass, obtained on DEXA evaluation, corrected by the body surface and that this index is a measure widely used in the assessment of muscle mass in the older adults (*). So, we decided to maintain the ASMi results in the present study.

Concerning the comment about the negative association between BMI, fat and vertical jumps, we would like to clarify that these findings are in agreement with a recent report published in 2020(**), in which the authors also demonstrated a negative correlation between fat mass and the performance of jump test. In addition, it is also worth mentioning that the scientific literature has been shown that excess body fat can negatively influence muscle performance(***). Thus, we added a new sentence concerning this issue in the "Discussion section", as presented below:

"These findings corroborate the recent report of Moore and collaborators [**], in which the performance of jump test was negatively associated with fat mass in older adults. "

*Cruz-Jentoft, A. J., Bahat, G., Bauer, J., Boirie, Y., Bruyère, O., Cederholm, T., ... & Zamboni, M. (2019). Sarcopenia: revised European consensus on definition and diagnosis. Age and ageing, 48(1), 16-31. DOI: 10.1093/ageing/afy169.

** Moore, B. A., Bemben, D. A., Lein, D. H., Bemben, M. G., & Singh, H. (2020). Fat mass is negatively associated with muscle strength and jump test performance. The Journal of frailty & aging, 1-5. Doi:10.14283/jfa.2020.11

***Liu, X., Hao, Q., Yue, J., Hou, L., Xia, X., Zhao, W., ... & Dong, B. (2020). Sarcopenia, obesity and sarcopenia obesity in comparison: prevalence, metabolic profile, and key differences: results from WCHAT study. The journal of nutrition, health & aging, 1-9. Doi: 10.1007%2Fs12603-020-1332-5

Regarding the comment about the association between the handgrip and vertical jump tests found in this study, we agreed that another study could be performed in order to verify whether one test can be useful to replace another, even understanding that each one involves a certain degree of coordination of different muscle groups.

3- Conclusions How has the safety of the test been evaluated? that is, what results give information to confirm that vertical jumping is a safe way to evaluate the elderly.

Authors´ response: We would like to thank you for the comment and, in order to respond to you, we would like to inform you that the test safety was verified in two ways: (1) there were no accidents (eg falls) and no incidents (discomfort or acute pain) during the performance of the physical tests; and (2) one week after the performance of the physical tests, by a telephone call, we not only provided the "feedback" concerning the results obtained in the performance of the tests but also we verified and confirmed the absence of any change in subjective physical discomfort after the tests. In addition, It is utmost of importance to mention that our results were in agreement with the other three studies[*] that reported not having had incidents and falls in the application of the vertical jump test. In this respect, and in order to be clearer, we decided to add the term "robust" to the title of our study due to the fact that older women who participated in our study were trained with vertical jump exercises.

Lastly, we would like to inform you that we added two sentences describing the procedure and the results regarding this evaluation in the "Methods" and "Results" sections, respectively, were added in the main text. These new sentences are presented below:

In the Methods section: "During the performance of the tests, the occurrence or not of falls, as well as the presence or absence of pain, and physical incidents were verified. In addition, one week after the second meeting, we contacted all volunteers, by telephone call, to provide feedback concerning the results of the assessments and also to verify the occurrence of any discomfort or pain that has been begun after the physical tests were performed."

In the Results section: "Concerning the evaluation of the safety of the physical tests applied in this study, it was not verified falls or physical accidents during their performances. In addition, by telephone contact one week following the tests performances, we verified that there was no occurrence of discomfort or pain related to the isokinetic evaluation and vertical jump tests.”

[*]

-Singh, H., Kim, D., Bemben, M. G., & Bemben, D. A. (2017). Relationship between muscle performance and DXA-derived bone parameters in community-dwelling older adults. Journal of musculoskeletal & neuronal interactions, 17(2), 50.

PMCID: PMC5492319

PMID: 28574411

-Alley, D. E., Shardell, M. D., Peters, K. W., McLean, R. R., Dam, T. T. L., Kenny, A. M., ... & Cawthon, P. M. (2014). Grip strength cutpoints for the identification of clinically relevant weakness. Journals of Gerontology Series A: Biomedical Sciences and Medical Sciences, 69(5), 559-566. DOI:10.1093/gerona/glu011

-Hicks, G. E., Shardell, M., Alley, D. E., Miller, R. R., Bandinelli, S., Guralnik, J., ... & Ferrucci, L. (2012). Absolute strength and loss of strength as predictors of mobility decline in older adults: the InCHIANTI study. Journals of Gerontology: Series A: Biomedical Sciences and Medical Sciences, 67(1), 66-73. DOI: 10.1093/gerona/glr055

Reviewer 3 Report

The study tries to present an evaluation of vertical jump tests in forty-one robust older women. Although the study intent to explain safety and physical performance assessment and the results could be interesting, the manuscript is not well-written, including the research question, while the methods section is not transparent sufficiently. The introduction lacks a scientific explanation of the study value. The study objective is not adequate. There is no proper discussion on the relevance of the study and the international implications in terms of its design, methods, results.  The remaining comments are highlighted in the document.

Round 2

Reviewer 1 Report

No comments

Author Response

Thank you very much for everything.

Reviewer 2 Report

Thank you for responding to each of the comments. I think the work has improved and is understood more clearly. 

Author Response

Thank you very much for the comments.

Reviewer 3 Report

The authors have put efforts to improve the manuscript according to the comments. The cover letter shows that they have addressed all comments.

I still have some concerns with the clarity of "safety", since that was not explained sufficiently. Is it a safe instrument, safe measurement, physical safety, safe environment or else? The term safety has rarely been mentioned. Let me help you understand my concerns: what would be "unsafe?", could you briefly explain that?

Suggestion: The responses in the cover letter can be used to strengthen the article description and value (i.e., regarding novelty, and regarding the relevance of tests).

Author Response

Comments of the Reviewer #3

Round #2

The authors have put efforts to improve the manuscript according to the comments. The cover letter shows that they have addressed all comments.

I still have some concerns with the clarity of "safety", since that was not explained sufficiently. Is it a safe instrument, safe measurement, physical safety, safe environment, or else? The term safety has rarely been mentioned. Let me help you understand my concerns: what would be "unsafe?", could you briefly explain that?

Suggestion: The responses in the cover letter can be used to strengthen the article description and value (i.e., regarding novelty, and regarding the relevance of tests).

Authors´ response: First of all, we would like to thank you for all your comments. In addition, we are grateful for your recommendation to clarify the meaning of "safety" in our study.

In this respect, we would like to inform you that the term "safety" in our study is closely associated with maintaining physical integrity aspects during the performance of the physical tests, especially to the vertical jump test, which by involving a complex motor activity, the occurrence of falls, pain, fractures, and other physical accidents can be reported. In addition, since "unsafe" can mean "dangerous" or "harmful", it is mandatory to avoid performing physical tests that could induce some physical accident, such as fall and/or musculoskeletal injury, particularly by older adults.

Lastly, in order to clarify the meaning of the term "safety" in the present study, we added pieces of information in the main text, marked in blue.
